# Soybean Response to Seed Inoculation or Coating with *Bradyrhizobium japonicum* and Foliar Fertilization with Molybdenum

**DOI:** 10.3390/plants12132431

**Published:** 2023-06-23

**Authors:** Wacław Jarecki

**Affiliations:** Department of Crop Production, University of Rzeszów, Zelwerowicza 4 St., 35-601 Rzeszów, Poland; wjarecki@ur.edu.pl

**Keywords:** *Glycine max* (L.) Merr., root nodule bacteria, nitrogen fixation, seed coating, foliar fertilization, molybdenum, yield, chemical composition

## Abstract

Soybean is one of the most important legumes in the world, and its advantages and disadvantages are well known. As a result of symbiosis with the bacterium *Bradyrhizobium japonicum*, soybean can assimilate nitrogen from the air and is therefore not fertilized with this element, or if it is, only at small doses. In soybean agriculture practice, an important treatment is the inoculation of seeds with symbiotic bacteria and optimal fertilization with selected nutrients. Therefore, a three-year (2019–2021) field experiment was carried out to investigate the effects of soybean in the field to a seed *Rhizobium* inoculation or coating and molybdenum foliar fertilization. There were no significant interactions between the tested treatments over the years. It was demonstrated that the best variant was seed inoculation before sowing in combination with foliar molybdenum application. As a result of this treatment, a significant increase in nodulation, soil plant analysis development (SPAD) index, leaf area index (LAI) and seed yield (by 0.61 t·ha^−1^) was obtained compared to the control. In addition, the content of total protein in the seeds increased, while the content of crude fat decreased, which significantly modified the yield of both components. Sowing coated seeds in the Fix Fertig technology was less effective compared to inoculation, but it was significantly better than that in the control. Coating seeds with *B. japonicum*, in combination with foliar fertilization with molybdenum, could be recommended for agricultural practice, which was confirmed by economic calculations. Future experiments will assess the soybean’s response to seed inoculation or coating and fertilization with other micronutrients.

## 1. Introduction

Soybean is the most important protein and oil plant in the world. It is cultivated on an area of 120–130 million hectares. The largest soybean producers are Brazil, the United States, Argentina and China [1]. In Europe, soybean meal and seeds are in high demand, but the region relies mainly on imports due to the small area of legume cultivation (only 1.5% of agricultural land) compared to the global needs (14.5%) [2]. Fortunately, in central Europe, it is possible to increase legume production, including that of new soybean varieties that are well adapted to the colder climate [3,4]. Jarecki and Migut [5] have confirmed that soybeans offer high and good-quality yields compared to other legumes. Prusiński et al. [6] have reported that interest in soybean cultivation in Poland is increasing, which results from climate change and economic factors. However, it is necessary to improve the agriculture practices of this species, especially seed inoculation and mineral fertilization to local habitats.

Zimmer et al. [3] and Narożna et al. [7] have shown that *B. japonicum* is not naturally present in European soils; thus, soybean seeds should be inoculated to increase root nodulation. However, commercial inoculants and mineral fertilization can be less effective under changing climatic conditions, according to studies by Prusiński et al. [6] and Ambrosini et al. [8]. Rainfall deficits have been shown to especially disrupt the nodulation process. Pannecoucque et al. [9] showed that experiments with seed inoculation were particularly important in regions with colder climates and when soybeans were sown for the first time in a rotation. Kühling et al. [10] confirmed that the efficiency of seed inoculation was lower when soybean was grown in a cold region, but this treatment was still necessary because the soils lacked *B. japonicum*. In this aspect, Albareda et al. [11] and Narożna et al. [7] proved that the introduction of symbiotic bacteria into the soil, where they are not present, resulted in their high abundance in subsequent years.

Thilakarathna and Raizada [12] have concluded that farmers are interested in bio-inoculants, particularly as the prices of mineral fertilizers increase and the need to reduce their impact on the environment becomes more pressing. Commercial inoculants or seed coating are a good example of this in the case of legumes. Duzan et al. [13] have pointed out, however, that legumes are sensitive to environmental stresses, resulting in variable yields over the years. For example, low or high temperature, lack or excess of water, high salinity or low pH adversely affect nodulation, which reduces the extent of biological nitrogenfixation (BNF) and the final yield. Salvagiotti et al. [14] showed that soybean’s nitrogen requirement was 50–60% covered by biological N_2_ fixation. This is particularly important information when high-yielding varieties with higher nutrient requirements are grown [15]. Cooper [16] reported that yields of the best soybean varieties were 5 t ha^−1^, and the yield potential was even 7 t ha^−1^. In this case, nodulation must be very efficient to meet the plants’ nitrogen requirements. Salvagiotti et al. [14] argued that when cultivating high-yielding varieties, fertilization with a small dose of nitrogen might be necessary. However, the main source of this element should be biologically fixed nitrogen. The experiments carried out by Jarecki [17] showed that seed inoculation had a positive effect on nodulation and physiological parameters of soybean plants. However, the farmer must carry out such treatment on his own farm, which is costly and time-consuming. Althabegoiti et al. [18] demonstrated that the application of symbiotic bacteria to the soil could be an alternative because the effects of such treatment were better than seed inoculation. Pedrini et al. [19] reported that some companies offered ready-for-sowing seeds coated with an appropriate bacterial strain; however, according to Wächter et al. [20], the effectiveness of coating could be lower than the traditional seed inoculation, although it saves time for farmers, especially for large soybean cultivations. Deaker et al. [21] found that the use of a higher dose of inoculant than recommended did not pose any threat to the environment and usually resulted in an increase in nodulation and seed yield by up to 25%. However, it should be noted that the inoculation of a large number of seeds is technically difficult and expensive. Carciochi et al. [22] reported that there is no benefit from using larger amounts of inoculants on soils where soybean is grown in good climatic conditions. López-García et al. [23] also showed that seed inoculation did not always result in an increase in yield or protein content in seeds, as habitat conditions and agricultural technology also played a significant role. Additionally, Bargaz et al. [24] and Beiranvand et al. [25] concluded that bio-inoculants could have a synergistic effect with mineral fertilization, depending on the element applied.

Many studies [26,27,28,29] have confirmed that the process of nodulation is affected by mineral fertilization, although not always positively. Brzezińska and Mrozek-Niećko [30] proved that a fertilizer containing copper and manganese exerted a toxic effect on symbiotic bacteria. According to Bagale, [31] soybean requires primarily nitrogen, phosphorus, potassium, calcium, magnesium and sulfur, and of micronutrients: iron, boron, zinc, cobalt, copper, manganese, molybdenum, nickel and chlorine. These nutrients are typically supplied in various fertilizers [32,33], with microelements preferably applied foliarly [34,35]. Thapa et al. [36] showed that foliar fertilization was more effective in high-yielding soybean varieties because it improved the physiological parameters of plants and canopy architecture. Therefore, they recommend to analyze the plants in conjunction with regular soil testing. Most studies evaluating soybean yield response to the foliar application of micronutrients have shown mixed results. Therefore, any additional foliar feeding should be considered under conditions that do not jeopardize the production cost. Banerjee and Nath [37] proved that molybdenum fertilization exerted a synergistic effect on biological nitrogen fixation (BNF), especially in acidic soils. Molybdenum (Mo) acts as a cofactor for nitrogenase and nitrate reductase enzymes, which are directly involved in the biological fixation and subsequent assimilation of nitrogen in legumes. Thus, Mo plays a crucial role in nitrogen metabolism and protein synthesis. It also facilitates diverse physiological and biochemical processes, including photosynthesis as well as carbohydrate and sulfur metabolisms. Weisany et al. [38] have confirmed that the role of molybdenum is particularly important in legumes, and it is best to use this element in foliar fertilizers. As molybdenum is a metal component of nitrogenase, all N2-fixing systems have a specific high molybdenum requirement. Molybdenum deficiency-induced nitrogen deficiency in legumes relying on N2 fixation is widespread, particularly in acid mineral soils.

In summary, soybean is an important crop that provides nutritional, economic, and soil fertility benefits for farmers [39]. Therefore, it is essential to improve agricultural practices and disseminate innovative solutions to enhance soybean cultivation.

The aim of the experiment was to investigate the response of soybean to seed inoculation with *B. japonicum* and foliar fertilization with molybdenum. The research hypothesis assumed that the effectiveness of nodulation would be higher after the inoculation of symbiotic bacteria in combination with the foliar application of molybdenum.

## 2. Results

### 2.1. Field Measurement Results

There were no significant interactions between the tested treatments over the years. The sowing of coated seeds (variants SC and SC + FF) resulted in the one-day-later emergence of plants. The earliest emergence of plants was recorded in 2021, while the latest was in 2020, and this difference was statistically significant. Plant density before harvest varied significantly only in the years of the study. The SPAD index measurements showed that the plants were most optimally nourished after seed inoculation in combination with foliar Mo fertilization. Significantly lower SPAD measurements were obtained on the control plot and when only coated seeds were sown or only when foliar fertilization was applied. LAI measurements showed a similar relationship. After the use of seed inoculation in combination with foliar fertilization, the plants developed the largest leaf area per 1 m^2^. Significantly lower LAI index measurements were recorded on the control plot and when only coated seeds were sown or only when foliar fertilization was applied (Table 1).

### 2.2. Nodulation

The number of nodules on the roots depended on the interaction of the tested factor with study years. The highest nodulation was recorded in 2020 after the application of seed inoculation in combination with foliar fertilization. Seed inoculation without molybdenum application also resulted in the development of a large number of nodules. In 2019, nodulation was the weakest. Although the number of nodules on roots was satisfactory, it was usually lower in the case of sowing coated seeds compared to the inoculation treatment. As expected, seed sowing without symbiotic bacteria resulted in sparse root nodules (Table 2).

There was no proven interaction between the tested factor and the years of research for nodule dry weight. Sowing of inoculated or coated seeds resulted in a significant increase in nodule dry weight compared to control. However, it should be noted that foliar application of molybdenum did not significantly increase nodule dry weight in variants SI + FF and SC + FF compared to variants SI and SC. Sowing seeds without root-nodule bacteria (variant C and FF) resulted in the lowest score for the trait tested (Figure 1).

### 2.3. Yield Components and Yield

The number of pods per plant significantly increased after seed inoculation together with foliar fertilization, but only in comparison to the control. In 2019 and 2021, this trait amounted to 28.29 and 34.96 pods, respectively. However, the number of seeds per pod was not modified by the factor studied or by the years of the study. TSW, on the other hand, significantly differed over the years of the study, with the largest seeds obtained in 2021. Seed yield was the highest after inoculation in combination with foliar fertilization. Significantly lower yields were obtained after sowing inoculated seeds but without molybdenum application, and when coated seeds were sown (regardless of foliar fertilization). Soybean yielded the lowest on the control plot and plots with only foliar Mo fertilization. The difference in yields between variant SI + FF and control C was 0.61 t ha^−1^ (Table 3). Soybean yields varied during the study years, with a difference of 1.35 t·ha^−1^ between 2021 and 2019. Thus, the weather conditions had a decisive influence on the soybean yields. However, there was no significant interaction between the tested factor and the years of research.

### 2.4. Chemical Composition of the Seeds

There was no significant interaction between the tested factor and the years of research. The protein and fat content in the seeds and the yield of both components were significantly modified (Table 4). Seed inoculation with or without Mo foliar fertilization (SI or SI + FF) significantly increased seed protein content compared to the control, but only variant SI + FF decreased the fat content. The lowest protein content was determined in the control seeds. Seed inoculation with foliar fertilization resulted in the highest protein yield, which was due to the combination of high seed yield and protein content. However, fat yield was also high in variant SI + FF, which was associated solely with high seed yield.

### 2.5. Economic Results

Economic calculations (Table 5) showed that the use of seed inoculation in combination with foliar Mo application provided the best result. It is important to note that seed inoculation must have been performed on the farm, which was a time-consuming procedure. Sowing of pelleted seeds was less profitable than inoculation, but still more advantageous compared to the control. Foliar fertilization with molybdenum was economically justified in both variants (SI + FF and SC + FF).

## 3. Discussion

Soybean is one of the world’s oldest and most valuable crop plants grown. It owes its popularity to its use value as well as its positive influence on soil and successive plants. In Europe, efforts have been made to increase soybean production through the use of optimal varieties and agricultural practices. My study showed that variable weather conditions significantly modified only the number of nodules on the soybean roots, as demonstrated by the FxY interaction (Table 2). However, the interaction was not statistically proven for the other traits and parameters studied. Therefore, the results obtained were analyzed for overall averages. Kuchlan et al. [40] showed that soybean was very sensitive to weather conditions and other environmental stresses, resulting in unstable yields over the years, which is characteristic of legumes. Macák and Candráková [41] proved that some biometric soybean traits, such as the number of pods per plant, depended on weather conditions, while others, e.g., the number of seeds per pod, did not.

This was also confirmed in my study (Table 3). Among the yield components, the number of seeds per pod did not vary across the study years, while plant density before harvest, the number of pods per plant and TSW differed over the years. It was shown that sowing coated seeds resulted in delayed plant emergence compared to the inoculated seeds and the control (Table 1), but the difference was not significant, averaging only one day. Sharratt and Gesch [42] cautioned against sowing coated seeds, especially under conditions of soil water shortages and high temperatures, as some coatings could delay the germination and emergence of soybean.

Vollmann et al. [43] reported that modern measurement methods of, e.g., chlorophyll content in leaves, allowed for the rapid assessment of plant nutritional status. The information obtained can help determine the availability of symbiotic nitrogen and the need for fertilization. Gwata et al. [44] have argued that if the symbiotic fixation of atmospheric nitrogen (N_2_) in the root nodules is adequate, then fertilization is not necessary. However, if the nodulation is weak, it may be necessary to apply a low dose of mineral nitrogen.

The SPAD index measurement performed in my study showed that the plants were most optimally nourished after applying seed inoculation together with foliar Mo fertilization. Significantly lower readings were obtained in the control and when only coated seeds were sown, or when only foliar fertilization was applied. The LAI measurement results showed a similar relationship (Table 1). Vollmann et al. [43] and Nasar et al. [34] reported that biological nitrogen fixation has a large impact on photosynthesis and other physiological processes of legumes. Therefore, the non-destructive measurement of the chlorophyll content in leaves can provide information on nodulation in early developmental stages. Thompson et al. [45] concluded that the assessment of the nutritional status of plants could be performed, e.g., using a Minolta 502 SPAD meter. Such a measurement is quick and easy to perform both in the laboratory and in field conditions. Kühling et al. [10] showed that the SPAD index was significantly higher when inoculated soybean seeds were sown, but only at the seed development stage. Measurements conducted in earlier developmental stages did not differ significantly from plants without inoculant. Wiatrak [46] proved that the coating of soybean seeds increased the NDVI and LAI index under drought stress conditions, resulting in significantly higher seed yields compared to the control group.

The present study showed that sowing inoculated seeds significantly increased the number of nodules on soybean roots compared to the control. The highest degree of nodulation was recorded in 2020 after seed inoculation in combination with foliar fertilization (Table 2). A high number of nodules developed also after seed inoculation without molybdenum foliar application. Sowing coated seeds (SC or SC + FF) increased the number of nodules on the roots, but in general, there were fewer of them compared to those with the inoculation procedure (SI or SI + FF). In addition, the results obtained were variable in the years of research. Seed-coating technology has developed rapidly during the past two decades and has provided an economical approach to seed enhancement, especially for larger seeded agronomic and horticultural crops, but my results show that traditional inoculation is more effective.

Solomon et al. [47] confirmed that seed inoculation had a beneficial effect on the number and dry weight of nodules and, as a result, on the yield components and soybean yield. Marinković et al. [48] reported that the inoculation procedure increased biological nitrogen fixation, which allowed for the proper growth and development of legumes, improved soil fertility, and reduced nitrogen fertilizer doses. In this aspect, Thilakarathna and Raizada [12] showed that nodulation was influenced by many factors, including soil humus content, fertilization, pH, salinity, agricultural practices, as well as temperature and drought. Kaschuk et al. [49] concluded that normal biological nitrogen fixation could meet the soybean demand for this element even in high-yielding varieties. On the other hand, other nutrients had to be provided with fertilizers, especially if they had a synergistic effect on nodulation [50]. The exceptions were copper and manganese, which limited the development of nodules on soybean roots [30]. Sogut [51] showed that seed inoculation with a low nitrogen dose had a positive effect on soybean yield. This was particularly evident in varieties with a long growing season. Gewehr et al. [52] proved that inoculation in combination with molybdenum fertilization resulted in better nitrate reductase activity and plant vigor, which improved the size and quality of the soybean yield.

In the present experiment, the sowing of inoculated seeds in combination with foliar Mo application resulted in a significant increase in the number of pods per plant compared to the control. The number of seeds per pod was not significantly modified, and TSW varied only in the years of the study. Vollmann et al. [43] confirmed that nodulation had a significant effect on plant biometric traits, including some yield components. Salvagiotti et al. [14] reported that soybeans showed clear symptoms of nitrogen deficiency, which affected plant growth and development. Therefore, Sessitsch et al. [53] and Leggett et al. [54] have argued that seed inoculation is an important procedure in agricultural practice, and research in this field should be continued, especially in new regions of soybean cultivation.

In my experiment, the combination of sowing inoculated seeds with foliar fertilization had the most beneficial effect on soybean yield and economic effects (Table 5). Significantly lower yields were obtained after sowing inoculated seeds without Mo application or when coated seeds were sown (regardless of foliar fertilization). Soybean yielded the lowest in the control without *B. japonicum* (Table 3). The difference in seed yield between variant SI + FF and control was 0.61 t·ha^−1^, while the difference in seed yield between 2019 and 2021 was 1.35 t·ha^−1^, demonstrating the strong influence of weather conditions on soybean yields. Economic calculations (Table 5) showed that the use of the SI + FF variant was justified, despite the costs.

Kühling et al. [10] reported that seed inoculation did not always increase the yield and quality of soybean seeds, particularly in colder regions or years. The increase in protein yield as a result of inoculation was confirmed only in one location. On the other hand, Jarecki and Wietecha [55] proved that sowing coated seeds did not always outperform the control, depending on variable weather conditions and coat composition. In other experiments, inoculating [11,29] or coating [56] seeds significantly increased soybean yield, especially when combined with optimal fertilization with macronutrients and micronutrients that have a synergistic effect on nodulation. Molybdenum [29,57], including its foliar application [58,59], is an example of a micronutrient that has a positive effect on the process of biological nitrogen fixation. The effects of molybdenum fertilization depend on the variety, region or year of research. Banerjee and Nath [37] demonstrated that Mo fertilization was effective in acidic soils lacking this element. Oliveira et al. [35] reported that the foliar application of Mo positively influenced physiological plant processes, resulting in higher yields. Bambara and Ndakidemi [60] reported that molybdenum was a component of some bacterial nitrogenases, and therefore is especially important for plants that live in symbiosis with nitrogen-fixing bacteria such as Rhizobium.

The current study revealed that seed inoculation together with foliar Mo fertilization significantly increased seed protein content but decreased the fat content. The lowest protein content was determined in the control seeds (Table 4). The protein and fat yield were the highest after seed inoculation in combination with foliar Mo application (variant SI + FF). At the same time, the fat yield was influenced by the high seed yield and not by the content of this component in the seeds. This was consistent with the study by Flajšman et al. [61], which indicated that the inoculation treatment significantly increased seed protein content and the protein and fat yield compared to control. Albareda et al. [11] and Vollmann et al. [43] also suggested an increase in soybean seed protein content with a higher number of root nodules. Cafaro La Menza et al. [62] reported that insufficient nitrogen availability for plants always resulted in a decrease in the seed, protein and fat yields. Therefore, research related to soybean inoculation and fertilization is considered important for agricultural practices [63].

## 4. Materials and Methods

This experiment was carried out in 2019–2021 in a field of the Podkarpackie Agricultural Advisory Centre PODR in Boguchwała (21°57′ E, 49°59′ N), Podkarpackie Province, Poland. The experiment was carried out in four replicates in a random block design. The tested factor was a bacterial vaccine for soybean (*Bradyrhizobium japonicum*) and foliar fertilizer with molybdenum applied in the following variants:C—Control (without inoculant and foliar fertilization);SI—Seeds inoculated before sowing;SC—Seeds inoculated (coated), Fix Fertig technology;FF—Foliar fertilization with molybdenum;SI + FF;SC + FF.

This experiment was carried out using the variety Abelina, recommended for cultivation in the study area. The commercial preparation HiStick^®^ Soy (BASF, Ludwigshafen am Rhein, Germany) was used for seed inoculation (variant SI), which was dry-mixed with the seeds directly before sowing. This product contains at least 2 × 10^9^ (at least 2 billion) viable Rhizobium (*Bradyrhizobium japonicum*) bacterial cells for use in soybean cultivation per gram of peat substrate. The original polymer was added to the peat substrate at a low concentration to ensure adhesion and safety.

In variant SC, the seeds were inoculated *B. japonicum* (commercially coated seeds) using the Fix Fertig technology (Saatbau Polska Sp. z o.o., Środa Śląska, Poland). This allowed for the obtaining of the so-called “ready-to-sow” seeds, which do not need to be inoculated before sowing. In this process the seeds are coated with rhizobia together with a polymer, which acts as a preservative and also protects against solar radiation. No chemical dressing was applied to the seeds.

Mikrovit^®^ Molybdenum (Intermag sp. z o.o., Olkusz, Poland), containing 33 g L^−1^ Mo, was selected for foliar fertilization. Application was conducted three times with a hand sprayer at the following BBCH stages: 12 (first trifoliate leaf), 51 (beginning of budding) and 71 (first pod development). The manufacturer’s recommendations were to use 1 L of the preparation and 300 L of water per hectare per treatment. Plant development stages were given according to the BBCH scale (Bundesanstalt, Bundessortenamtund CHemische Industrie) [64].

This experiment was established on sandy loam soil, Haplic Luvisol [65], which was slightly acidic, ranging from 5.8 to 6.3 mol/L KCl. The content of available phosphorus (P_2_O_5_ from 16.9 to 18.8 mg·100 g^−1^ of soil), potassium (K_2_O from 19.8 to 21.7 mg 100 g^−1^ of soil) and magnesium (Mg from 5.6 to 6.5 mg 100 g^−1^ of soil) were very high or high. The determined micronutrient contents were average (Table 6). Soil sample analyses were carried out at the District Chemical-Agricultural Station in Rzeszów following Polish standards. It should be noted that soil molybdenum analyses are not standard procedures in Poland.

The weather conditions during this experiment were recorded using data from the meteorological station of the University of Rzeszów located in the Rzeszów Zalesie Municipal District, 5 km away from the experimental field.

The weather conditions varied in the years of the study, which affected some of the parameters examined. The highest rainfall in the analyzed months was recorded in 2021, and the lowest in 2019. High precipitation was recorded in June 2020 and August 2021. April, June and August of 2020 were dry months (Table 7). The warmest month was June 2019 and July 2021. On the other hand, May in 2020 and April and August in 2021 were cold.

Seed sowing in individual years was performed on 16 April 2018, 15 April 2019, and 21 April 2020. The plot area was 15 m^2^ with 1.5 m^2^ insulation strips. Soybean was not grown in the experimental field beforehand, and winter wheat was the forecrop. Sixty germinating seeds were sown per 1 square meter. Row spacing was 45 cm, and sowing depth was 3.5 cm. Mineral fertilization was as follows: 40 kg ha^−1^ P_2_O_5_ (superphosphate 19%) and 60 kg ha^−1^ K_2_O (potassium salt 60%), while mineral nitrogen fertilization was not applied.

Mandryl 500 SC (metobromuron) and Corum 502.4 SL (bentazone, imazamox) were used for weed control, while no insecticides or fungicides were applied.

At the BBCH 79 stage (formed pods), 10 roots were dug out and subsequently rinsed on sieves, and the number of nodules was counted. The dry weight of nodules was subsequently measured after drying at 20 °C.

Soil plant analysis development (SPAD) and leaf area index (LAI) measurements were performed at the BBCH 79 stage. A SPAD 502P chlorophyll meter (Konica Minolta, Inc., Tokyo, Japan) was used for SPAD index measurements. Leaf area index (LAI) measurements were performed using an AccuPAR LP-80 apparatus (Meter Group, Inc., Pullman, WA, USA).

Biometric measurements (number of pods per plant, number of seeds per pod) were carried out on 20 plants collected from plots at the technical maturity stage (BBCH 96). Thousand-seed weight (TSW) was determined with one decimal place precision. Plant density before harvest was calculated per 1 m^2^. Soybean was harvested at full maturity, and seed yield from the plots was calculated per 1 ha, taking into account 14% moisture.

The chemical composition of seeds (total protein, crude fat) was determined by the near-infrared method using an FT-LSD MPA spectrometer (Bruker, Germany) in the laboratory of the Department of Plant Production, University of Rzeszów. FT-NIR technology measures the absorption of the near-infrared light of the sample at different wavelengths. The components yield per ha was calculated based on the seed yield and protein or fat percentage.

Prices for economic calculations were based on those in 2023, in line with the offer of commercial companies and available data from the agricultural market. The average of the study years was used as seed yield. The exchange rate used was EUR 1 = PLN 4.74, and the purchase price of seeds was EUR 527.4 per ton. The cost of foliar fertilizer was 5.8 EUR/liter, and spraying was EUR 10.5 per 1 ha. The cost of the inoculant was EUR 26.58, and the labor was EUR 4.22 per hour. Coated seeds were EUR 21.1 more expensive per ha compared to non-coated seeds.

The results were statistically analyzed with the analysis of variance (ANOVA), and Tukey’s half-confidence intervals were used to determine the significance of differences between the characteristic values. Statistical analysis was performed using TIBCO Statistica 13.3.0 (TIBCO Software Inc., Palo Alto, CA, USA).

## 5. Conclusions

Seed inoculation with symbiotic bacteria, especially in soils lacking them naturally, and optimal nutrient fertilization are considered particularly important treatments in soybean cultivation. The present study examined the reaction of soybean to seed inoculation or coating with *Bradyrhizobium japonicum* and the synergistic effect of foliar fertilization with molybdenum. It was demonstrated that seed inoculation combined with foliar molybdenum application significantly increased nodulation on roots, SPAD index, LAI index, and seed protein content, as well as seed, protein and fat yields compared to the control, with the only reduction being in crude fat content. The difference in seed yield between variant SI + FF and control was 0.61 t·ha^−1^, and sowing seeds coated with Fix Fertig technology was less effective compared to sowing seeds with inoculation, but significantly better than the control results. Foliar fertilization with molybdenum was less effective than expected, but it was still economically viable. Soybean yielded variably during the study years, and the difference obtained between 2021 and 2019 was 1.35 t·ha^−1^. Future experiments will explore the response of soybean to seed inoculation/coating in combination with fertilization using other micronutrients.

## Figures and Tables

**Figure 1 plants-12-02431-f001:**
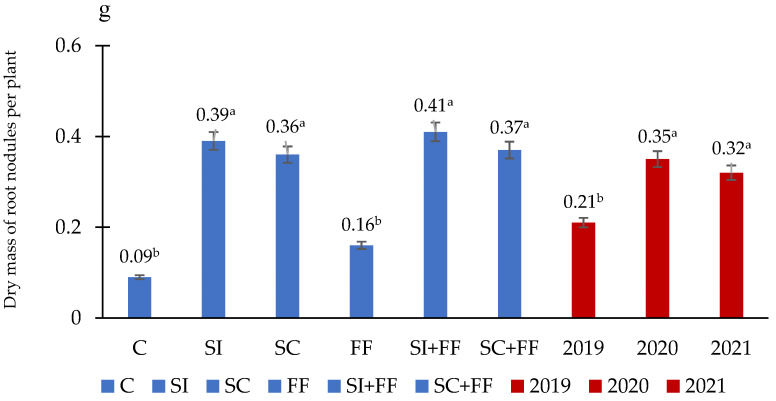
Dry mass of root nodules per plant. C—control (without inoculant and foliar fertilization), SI—seeds inoculated before sowing, SC—commercially coated seeds—Fix Fertig technology, FF—foliar fertilization with molybdenum, SI + FF, SC + FF. Different letters in the same column indicate significant differences (*p* < 0.05) according to the analysis of variance (ANOVA).

**Table 1 plants-12-02431-t001:** Field measurements.

Specification	Emergence Phase (Days from Sowing)	Plant Density before Harvest (pcs·m^−2^)	SPAD	LAI (m^2^/m^2^)
Factor (F)				
C	11.2 ± 2.12 b	43.58 ± 4.76 a	39.08 ± 2.94 b	3.94 ± 0.66 b
SI	11.2 ± 1.90 b	44.75 ± 5.05 a	41.17 ± 2.48 ab	4.15 ± 0.55 ab
SC	12.2 ± 2.29 a	44.33 ± 4.87 a	39.92 ± 2.31 bc	3.97 ± 0.65 b
FF	11.3 ± 2.14 b	44.08 ± 5.33 a	40.08 ± 3.21 bc	3.96 ± 0.66 b
SI + FF	11.2 ± 1.85 b	44.92 ± 4.23 a	42.08 ± 2.39 a	4.22 ± 0.55 a
SC + FF	12.2 ± 2.21 a	44.58 ± 5.38 a	40.75 ± 2.34 abc	4.03 ± 0.65 ab
Years (Y)				
2019	11.67 ± 1.05 b	43.25 ± 2.21 b	38.54 ± 1.77 b	3.55 ± 0.18
2020	13.75 ± 0.79 a	39.67 ± 2.06 c	43.42 ± 1.69 a	3.73 ± 0.21
2021	9.13 ± 0.54 c	50.21 ± 1.38 a	39.58 ± 1.74 b	4.85 ± 0.19
Interaction				
F x Y	r.n.	r.n.	r.n.	r.n.

C—control (without inoculant and foliar fertilization), SI—seeds inoculated before sowing, SC—commercially coated seeds—Fix Fertig technology, FF—foliar fertilization with molybdenum, SI + FF, SC + FF. Results are expressed as mean value ± standard deviations. Different letters in the same column indicate significant differences (*p* < 0.05) according to the analysis of variance (ANOVA).

**Table 2 plants-12-02431-t002:** The influence of the interaction of the tested factor and the years of research on the number of nodules on the root per plant.

Treatments	2019	2020	2021	Mean
C	0.56 g	0.84 g	0.69 g	0.70
SI	16.03 cde	20.7 ab	17.45 abc	18.06
SC	11.28 f	16.45 cd	12.95 def	13.56
FF	0.57 g	0.87 g	0.72 g	0.72
SI + FF	17.3 bc	21.45 a	18.2 abc	18.98
SC + FF	12.28 ef	17.45 abc	14.95 cdef	14.89
Mean	9.67	12.96	10.83	11.15

C—control (without inoculant and foliar fertilization), SI—seeds inoculated before sowing, SC—commercially coated seeds—Fix Fertig technology, FF—foliar fertilization with molybdenum, SI + FF, SC + FF. Different letters in the same column indicate significant differences (*p* < 0.05) according to the analysis of variance (ANOVA).

**Table 3 plants-12-02431-t003:** Yield components and seed yield.

Specification	The Number of Pods on the Plant	The Number of Seeds in the Pod	Thousand Seed Weight (g)	Seed Yield (t·ha^−1^)
Factor (F)				
C	29.75 ± 3.41 b	1.95 ± 0.05 a	131.7 ± 9.75 a	3.34 ± 0.62 d
SI	31.92 ± 3.70 ab	1.98 ± 0.07 a	130.8 ± 10.59 a	3.70 ± 0.64 b
SC	31.08 ± 3.68 ab	1.97 ± 0.07 a	130.6 ± 10.64 a	3.54 ± 0.60 bc
FF	30.67 ± 3.70 ab	1.99 ± 0.06 a	130.2 ± 10.97 a	3.49 ± 0.64 cd
SI + FF	33.42 ± 3.39 a	2.01 ± 0.07 a	131.5 ± 10.24 a	3.95 ± 0.61 a
SC + FF	32.00 ± 3.49 ab	1.98 ± 0.08 a	132.1 ± 9.96 a	3.74 ± 0.72 b
Years (Y)				
2019	28.29 ± 2.93 c	2.00 ± 0.06 a	124.0 ± 3.94 b	3.03 ± 0.40 c
2020	31.17 ± 1.93 b	1.96 ± 0.05 a	143.5 ± 5.06 a	3.48 ± 0.23 b
2021	34.96 ± 2.26 a	1.98 ± 0.07 a	125.9 ± 5.31 b	4.38 ± 0.32 a
Interaction				
F x Y	r.n.	r.n.	r.n.	r.n.

C—control (without inoculant and foliar fertilization), SI—seeds inoculated before sowing, SC—commercially coated seeds—Fix Fertig technology, FF—foliar fertilization with molybdenum, SI + FF, SC + FF. Results are expressed as mean value ± standard deviations. Different letters in the same column indicate significant differences (*p* < 0.05) according to the analysis of variance (ANOVA).

**Table 4 plants-12-02431-t004:** The content of protein and fat in seeds and the yield of both components.

Specification	Protein (% DM)	Fat (% DM)	Protein Yield (t·ha^−1^)	Fat Yield (t·ha^−1^)
Factor (F)				
C	37.93 ± 0.76 ^c^	19.38 ± 0.27 ^ab^	1.27 ± 0.26 ^c^	0.65 ± 0.11 ^c^
SI	38.93 ± 0.76 ^ab^	19.16 ± 0.25 ^bc^	1.44 ± 0.28 ^ab^	0.71 ± 0.12 ^abc^
SC	38.48 ± 0.79 ^bc^	19.25 ± 0.26 ^ab^	1.37 ± 0.26 ^bc^	0.68 ± 0.11 ^bc^
FF	38.13 ± 0.71 ^bc^	19.49 ± 0.28 ^a^	1.33 ± 0.27 ^bc^	0.67 ± 0.12 ^bc^
SI + FF	39.23 ± 0.83 ^a^	18.90 ± 0.33 ^c^	1.55 ± 0.27 ^a^	0.74 ± 0.11 ^a^
SC + FF	38.65 ± 0.86 ^abc^	19.10 ± 0.30 ^bc^	1.45 ± 0.31 ^ab^	0.71 ± 0.13 ^ab^
Years (Y)				
2019	37.85 ± 0.63 ^c^	19.40 ± 0.36 ^a^	1.15 ± 0.16 ^c^	0.59 ± 0.07 ^c^
2020	38.38 ± 0.59 ^b^	19.26 ± 0.29 ^a^	1.33 ± 0.11 ^b^	0.67 ± 0.04 ^b^
2021	39.45 ± 0.53 ^a^	18.99 ± 0.19 ^b^	1.73 ± 0.14 ^a^	0.83 ± 0.06 ^a^
Interaction				
F x Y	r.n.	r.n.	r.n.	r.n.

C—control (without inoculant and foliar fertilization), SI—seeds inoculated before sowing, SC—commercially coated seeds—Fix Fertig technology, FF—foliar fertilization with molybdenum, SI + FF, SC + FF. Results are expressed as mean value ± standard deviations. Different letters in the same column indicate significant differences (*p* < 0.05) according to the analysis of variance (ANOVA).

**Table 5 plants-12-02431-t005:** Economic effects.

Factor	Mean Yield (t·ha^−1^)	Mean Yield (EUR·ha^−1^)	Foliar Fertilization Cost (EUR·ha^−1^)	Inoculation Cost (EUR·ha^−1^)	Economic Result (EUR·ha^−1^)
C	3.34	1761.52	-	-	1761.52
SI	3.70	1951.38	-	30.80	1920.58
SC	3.54	1867.00	-	21.10	1845.90
FF	3.49	1840.63	48.90	-	1791.73
SI + FF	3.95	2083.23	48.90	30.80	2003.53
SC + FF	3.74	1972.48	48.90	21.10	1902.48

C—control (without inoculant and foliar fertilization); SI—seeds inoculated before sowing; SC—commercially coated seeds; Fix Fertig technology, FF—foliar fertilization with molybdenum (3 L per ha), SI + FF, SC + FF.

**Table 6 plants-12-02431-t006:** Chemical analysis of soil (30 cm).

Parameter	Unit	Year		
		2019	2020	2021
pH in 1 mol/L KCl	-	6.3	6.1	5.8
N_min_	kg∙ha^−1^	68	74	71
Humus	%	1.4	1.1	1.2
K_2_O	mg·100 g^−1^ soil	21.7	20.2	19.8
P_2_O_5_		18.8	17.5	16.9
Mg		6.5	5.9	5.6
Fe	mg·kg^−1^ soil	2015.3	2183.2	2415.4
Mn		221.6	308.8	345.4
Zn		11.8	12.4	12.8
Cu		4.6	5.8	5.9
B		1.1	1.3	0.9

**Table 7 plants-12-02431-t007:** Weather conditions.

Month	Sum of Precipitation (mm)			Temperature (°C)		
	2019	2020	2021	2019	2020	2021
April	21.4	10.0	49.4	9.9	9.2	6.5
May	73.5	83.3	63.9	13.1	11.3	12.8
June	30.8	162.9	47.3	21.5	18.1	18.8
July	49.8	18.9	55.0	19.1	18.8	21.6
August	60.9	7.3	107.4	20.3	19.9	17.5
September	32.0	44.1	85.9	14.7	15.0	13.1
Sum/Mean	268.4	326.5	408.9	16.4	15.4	15.1

## Data Availability

The data presented in this study are available on request from the corresponding author.

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
