# Peer review of "Soybean Response to Seed Inoculation or Coating with Bradyrhizobium japonicum and Foliar Fertilization with Molybdenum"

_plants, 2023, doi:10.3390/plants12132431_

Round 1
Reviewer 1 Report (Previous Reviewer 2)
Dear Authors,
After this second manuscript I understand better the treatments tested (my first demand) but some clarification is still required between SI and SC. I understand now that SI “seeds inoculated before sowing” comes from a BASF bio-inoculant; describe better the content of this product (strain of the Bradyrhizobium japonicum) and the proportion between the seeds and the bio-inoculant. For SC, we need to know if the only difference with SC comes from the seed coating with another commercial product, i.e. the Fix Fertig. Tell if you used the same inoculated seeds that for SC and describe also the proportion of the Fix Fertig with the seeds and its composition.
Another clarification which was asked during the first reviewing is to improve results’ presentation. I think that this could be also improved in homogenizing. For each variable, I think it is better to state firstly on each the treatements' factor and the years' factor interaction. A sentence like “there were no significant interactions between the tested treatments ant the years”. This introduction will allow you to present after each treatment effect. According to scientific standards, ”years” and “treatments” are factors with respectively three and six modalities. So, “a bacterial vaccine” is not a factor but ovementa bio-inoculant based on a BR strain.
Lastly, tell clearly that the experiment was carried-out during three years in the abstract. The two previous improvements need also to be applied in this part.
The conclusion needs to be improved as the first sentence should response directly to the tittle. Your first five lines are according to scientific standards, generalities, which generally are located in the introduction.
Regards.
See above.
Author Response
Please see the attachment

Reviewer 2 Report (Previous Reviewer 1)
The article has been improved significantly. The only issue I still have with it is Figure 1. I encourage the author to change the figure to a table. The author has already agreed to do so in response to my first review of the article. A table will significantly improve the results of the study.
Author Response
Please see the attachment

Reviewer 3 Report (New Reviewer)
The author described the effects of Bradyrhizobium japonicum and molybdenum foliar fertilization on soybean growth and production based on evidence from field experiments. However, it seems that the content of this manuscript is not qualified as a research article. More experiments should be conducted to study the potential mechanism of molybdenum foliar fertilization facilitates the soybean- B. japonicum complex. Otherwise, I suggest this manuscript is submitted as a communication.
In addition, there are some improvements could be done:
1. The revision of the MS is still tracked in the downloaded document. Is this the final version of the MS?
2. The meaning of C, SI,...should be given in the Table 1 rather than Table 4.
3. What does the "grey bar" mean in the error bar?
4. What are variants E and F in 2.2?
5. The subtitle of the discussion seems elusive, and there is no necessary to disscuss every aspect of your results individually. The discussion should be reorganized and the significance of this MS should be strengthened.
Minor editing of English language required
Author Response
Please see the attachment

Reviewer 4 Report (New Reviewer)
Manuscript ID: plants-2429832
Title: Soybean Response to Seed Inoculation or Coating with Bradyrhizobium japonicum and Foliar Fertilization with Molybdenum.
Author: Wacław Jarecki
We need to increase crop yields and produce more food for the growing population. In this study, the author explored the differences in some crop parameters (yield, protein composition) when different strategies for nodulation and fertilization are used, specifically molybdenum, across different growing seasons.
COMMENTS
I received a document that seems to be already reviewed and edited. Initially, I thought that the writing in some sections could be improved, but then I realized it looks more as the author's style, so I will not go there.
During the introduction, the author discussed the relevance of molybdenum during nodulation. However, it would be better to discuss the relevance of Mo in the aerial part of the plant since the strategy is to apply this micronutrient on the foliage and hence the enzymatic machinery that uses Mo.
There is no major effect on nodulation (number, weight) due to the different variants, except that SI performs a little better than the SC variants.
On page 4, second paragraph it is mentioned: “in variants E and F compared to variants SI and SC”. It should be FF variants.
Considering the stats, yield data did not show a clear benefit in all conditions tested. I think the text should reflect more on this. When the chemical composition is analyzed, there is an effect on the year of harvest, which should be discussed better together with the chemical properties of the soil, shown at the end of the paper. If the soil analysis is included earlier in the results section, it will improve the discussion. The acidification in soil over the years and the bioavailability of other micronutrients under this condition should play a role in some of the observed results. How much Mo is present in the soil?
SI variant with or without Mo foliar application offers a slight but better effect on the composition of seeds (protein and fat) but not the rest. Section 2.4 must describe Table 3 better.
In section 2.5, the author discussed that while SI + FF provides a better result, SC +FF would be more beneficial for the farmer (effort/cost). I would argue that the SI variant alone would give a similar benefit than SC + FF for less cost and effort.
Review the Discussion section, and I consider not including the role of the Mo FF on nodulation without discussing proper Mo mobilization mechanisms. I suggest focusing on the effect of Mo foliar application in the shoot instead of the root (except Mo soil data is available and included in the paper).
On page 9 paragraph 6, correct “ …of soil) was very high or high. “
Overall, this paper offers some clues on the use of different strategies to improve legume crop yield. There is still some more research needed to be done at molecular level to understand Mo fertilization alongside other micronutrients.
I would suggest this paper is accepted upon revision and changes I suggested are included.
Author Response
Please see the attachment

Reviewer 5 Report (New Reviewer)
Waclaw Jarecki described the effect of rhizobia inoculation and molybdenum fertilization on soybean in field condition. The results showed that there exists cooperated effect between rhizobia inoculation and molybdenum fertilization, which increases yield, LAI, content of protein and fat, economic result of soybean.
Major revision:
From the results, there had no more difference among molybdenum fertilization (FF), seeds inoculation (SC) and SC+FF, but SI+FF showed high difference with SI and FF. The author need make deep discussion about that.
Minor revision:
1. “symbiotically fixed nitrogen (BNF)”. It is wrong term. Symbiotic nitrogen fixation (SNF) and Biological nitrogen fixation (BNF).
2.The Y axle is not clear in Fig 1. “the number of nodules on the root” may describe as “the number of nodules on the root per plant”. Fig 2 is the same, missing “per plant”
Waclaw Jarecki described the effect of rhizobia inoculation and molybdenum fertilization on soybean in field condition. The results showed that there exists cooperated effect between rhizobia inoculation and molybdenum fertilization, which increases yield, LAI, content of protein and fat, economic result of soybean.
Major revision:
From the results, there had no more difference among molybdenum fertilization (FF), seeds inoculation (SC) and SC+FF, but SI+FF showed high difference with SI and FF. The author need make deep discussion about that.
Minor revision:
1. “symbiotically fixed nitrogen (BNF)”. It is wrong term. Symbiotic nitrogen fixation (SNF) and Biological nitrogen fixation (BNF).
2. The Y axle is not clear in Fig 1. “the number of nodules on the root” may describe as “the number of nodules on the root per plant”. Fig 2 is the same, missing “per plant”
Round 2
Reviewer 3 Report (New Reviewer)
Although this manuscript has been revised, the content of the manuscript is not significantly improved to qualify the journal as I suggested in the last review.
This manuscript is a resubmission of an earlier submission. The following is a list of the peer review reports and author responses from that submission.
Round 1
Reviewer 1 Report
Table 1: This table shows sum of precipitation and mean temperatures in 2019, 2020, and 2021. What does multiyear precipitation and temperature actually mean? Obviously, they are not means of the same month in the three years of the study. The multiyear data is confusing.
Figure 1: This is a very important figure of this article but it is difficult to comprehend. The figure shows the treatments, A-F, in the 2019, 2020, and 2021 study. The figure further says that different letters in the same column are statistically significantly different. Take the A-treatment as an example, the letter g indicates that treatment-A is not significantly different in each of the three years. But look at treatment-B, what do the letters cde in 2019, or ab in 2020 or abc in 2021 mean? What is cde in 2019 for one treatment (Treatment B) compared to? It is understandable to compare Treatment-B to itself in the different years (2019, 2020 and 2021), how do you explain comparing a treatment to itself in the same year? Also, what is column on a bar graph? Columns are best referred on tables not of graphs. This table needs to be rewritten for ease of comprehension. Also, there is no Y-axis label on the graph.
Chemical composition of the seeds: What was the method used to determine fat and protein contents of the seeds? This article says "Seed inoculation with foliar fertilization resulted in the highest protein yield, which was due to the combination of high seed yield and protein content. However, fat yield was also high in variant E, which was associated solely with high seed yield." How did you evaluate protein and fat? Did you use equal weight of seeds from every treatment for protein and fat evaluation? If not, you cannot conclude as stated in the article. Detail protein and fat assay and statistical analyses are required for each treatment to draw an unbiased conclusion of the effect of Treatment-E on protein and fat synthesis.
The article is nicely written in fairly good English language but the article has only one author. If this is the case, why is a collective pronoun used throughout the article. '........confirmed our study......" "......in our study......" One person cannot be referred to as "our".
Reviewer 2 Report
Dear authors,
A strong clarification is needed among the treatments tested. I suggest to put clearly the three (?) factors tested. Each factor has two modalities (without and with). The best design would be to test a full 3-factor factorial with eight treatments. As you have only six treatments, it is an uncomplete 3-factor design. The treatment/factor “C” Commercially coated seeds, Fix Fertig technology, is unclear. What is adding ?
Tell clearly in the abstract that you chose the statistical management of these data in considering one factor (six treatments) and to test the treatments*year interaction (three years).
Abstract
“… the reaction of soybean to the sowing of inoculated or coated seeds and foliar fertilization with molybdenum”: not clear.
I suggest: “ … the effects in field of soybean to a seed rhizobium inoculation, seed fertilizer coating and molybdenum foliar fertilization”.
“It was demonstrated that the best variant was seed inoculation before sowing in combination with foliar molybdenum application. As a result of these treatments”: not clear; “the best” is singular and “these” is plurial.
“ … to control”: to the control.
“ Fix Fertig technology”: not clear; describe what it contains.
Introduction
The introduction needs clearly to be separated in different paragraphs, each with a specific topic.
“average”: replace by “needs”, “individual” by “local”, “microbiological preparations” by “bio-inoculants”; “seed coats” by “seed fertilizer coating”.
“The aim of the experiment was to investigate the response of soybean to seed inoculation or seed coating with B. japonicum and foliar fertilization with molybdenum. The research hypothesis assumed that the effectiveness of nodulation would be higher after the application of symbiotic bacteria (inoculation, coating) in combination with foliar ap-plication of molybdenum”: not clear.
Results
Table 1 needs to be located in Material and Methods. I suggest a figure in transforming rainfall data in bars, with soybean cropping in lines.
This sentence “There was no significant interaction between the tested factor and the years of research” for plant variables, nodule dry weight and yield and its components, chemical composition of the seed needs to be the first sentence of the 2.2, 2.3, 2.4 parts.
Discussion
The discussion could be structured in sub-section 3.1, 3.2, each with a specific topic beginning with your main hypothesis, seed rhizobium inoculation.
The discussion needs to begin with a topic sentence linked with your tittle.
“Variable weather conditions significantly modified the number of nodules on soy-bean roots, as demonstrated by the FxY interaction”: no; you stated in the Results part that it is not significant.
Material and Methods
“The tested factor”. It is unclear. It seems that three factors are tested: seed bio-inoculation, fertilization seed coating (?) and molybdenum foliar fertilization. Further, we need to know what Fix Fertig technology contains and provides.
Treatments' abbreviation: instead of ABCDEF, letters could suggest the treatments and would make easier the reading e.g.: C or 0 as the control, BI as the Bio-inoculant, etc ...
Regards
Globally, easy ro read.